# Parallel Emergence of a Compartmentalized Striatum with the Phylogenetic Development of the Cerebral Cortex

**DOI:** 10.3390/brainsci9040090

**Published:** 2019-04-19

**Authors:** Tadashi Hamasaki, Satoshi Goto

**Affiliations:** 1Department of Neurosurgery, Kumamoto University Medical School, Kumamoto 860-8556, Japan; thamasaki-nsu@umin.ac.jp; 2Department of Neurodegenerative Disorders Research, Institute of Biomedical Sciences, Graduate School of Medical Sciences, Tokushima University, Tokushima 770-8503, Japan; 3Parkinson’s Disease and Dystonia Research Center, Tokushima University Hospital, Tokushima University, Tokushima 770-8503, Japan

**Keywords:** compartmentalization, matrix, striosome, phylogenetic development: striatum, vertebrate

## Abstract

The intricate neuronal architecture of the striatum plays a pivotal role in the functioning of the basal ganglia circuits involved in the control of various aspects of motor, cognitive, and emotional functions. Unlike the cerebral cortex, which has a laminar structure, the striatum is primarily composed of two functional subdivisions (i.e., the striosome and matrix compartments) arranged in a mosaic fashion. This review addresses whether striatal compartmentalization is present in non-mammalian vertebrates, in which simple cognitive and behavioral functions are executed by primitive sensori-motor systems. Studies show that neuronal subpopulations that share neurochemical and connective properties with striosomal and matrix neurons are present in the striata of not only anamniotes (fishes and amphibians), but also amniotes (reptiles and birds). However, these neurons do not form clearly segregated compartments in these vertebrates, suggesting that such compartmentalization is unique to mammals. In the ontogeny of the mammalian forebrain, the later-born matrix neurons disperse the early-born striosome neurons into clusters to form the compartments in tandem with the development of striatal afferents from the cortex. We propose that striatal compartmentalization in mammals emerged in parallel with the evolution of the cortex and possibly enhanced complex processing of sensory information and behavioral flexibility phylogenetically.

## 1. Introduction

The striatum is a key component of the basal ganglia circuit, which is involved in normal motor, cognitive, and emotional functions. Dysregulation of striatal function contributes to a variety of neurological and neurobehavioral disorders [1]. Accurate processing in the basal ganglia circuit is thus required for normal movements and behaviors in humans and depends on a balance between input and output neurotransmission, which in turn is reinforced by an organized cellular architecture [1]. The striatum is the primary receptive nucleus of the basal ganglia circuit and receives inputs from all cortical areas. However, the structural organization of the mammalian striatum is unique and is characterized by a mosaic pattern, distinct from the laminar cerebral cortex [2,3]. Two neurochemically-specified subdivisions, i.e., striosomes (patches) and the extrastriosomal matrix, exist in the mammalian striatum. This novel arrangement is related to neurotransmitter interactions and striatal input and output systems. The striosome is organized in a labyrinthine structure in 3D space, and the matrix occupies the space surrounding the striosome. The matrix-based output pathway acts as a push-pull system to increase or decrease movement through direct or indirect pathways, respectively [1]. The striosomal pathway likely exerts critical motor control by modulating nigral dopaminergic outputs [2,4]. Growing evidence suggests that the dysregulation of striatal compartments might cause movement disorders [1,5,6,7].

Little is known about striatal compartmentalization in non-mammalian vertebrates with primitive motor functions and behaviors. In this review, we focus on whether the mosaic organization of the striatum is unique to animals with well-developed cerebral cortices. We postulate that clear compartmentalization in the striatum emerges parallel to the phylogenetic development of the cerebral cortex. Do neurochemically-identified striosome and matrix neurons exist in non-mammalian vertebrates or anamniotes? Neurochemical markers to distinguish striosome and matrix compartments were extensively reviewed by Crittenden and Graybiel [1]. For example, D1-type dopamine receptor (D1R), glutamate receptor, ionotropic, AMPA 1 (GluR1), substance P (SP), and tyrosine hydroxylase (TH) in dopaminergic afferent terminals were markers for the striosome. Acetylcholine esterase (AChE), calbindin (CB), choline acetyltransferase (ChAT), enkephalin (ENK), and neuropeptide Y (NPY) were markers for the matrix. Here, we review the literature on expression patterns of striatal compartment markers throughout the vertebrate phylogeny (for reference see, Figure 1). Evolutional changes in fundamental architectures in the striatum were extensively addressed by Reiner et al. [8] and Smeets et al. [9] and are not presented in this review.

## 2. The Striatum in Anamniotes

Although the structure of the basal ganglia in early vertebrates has been extensively studied [12,13,14,15], little is known about striatal compartmentalization. Major neurotransmitters such as acetylcholine and dopamine, neuropeptides such as SP and ENK, and related enzyme systems such as AChE or TH, developed very early in animal evolution [8]. Their expression patterns can thus be used to study structural differences of the basal ganglia in diverse living anamniote species.

In lamprey, a very ancient anamniote, subpopulations of calbindin-negative striatal neurons projecting to structures other than the globus pallidus internus (GPi)/globus pallidus externa (GPe) and CB-positive neurons projecting to the GPi/GPe [13] are present. The striatum also receives dopaminergic innervation from the nucleus of the posterior tectum, a homolog of the substantia nigra pars compacta (SNc), suggesting that reciprocal connectivity between the striatum and the SNc was conserved in the oldest group of anamniotes [14]. Although neurochemical characteristics and connectivities of two distinct neuronal subpopulations in lamprey are similar to those of striosome and matrix neurons in mammals, clear demarcation into striosome and matrix neuron islands is absent [13,14].

Previous reports demonstrated the presence of a nucleus in the ventral telencephalon composed of neurons positive for SP [16] and GABA [17] in goldfish, and argued that it corresponds to the striatum [18]. Another study found the presence of an ascending dopaminergic system in zebrafish [19]. By investigating expression patterns of transcription factors that label neurons of the basal ganglia in adult zebrafish [20], one study showed that territories of the striatum and pallium were distributed in the rostro-caudal axis of the basal telencephalon. The location, neurochemical properties, and connectivity of these neuronal populations suggested a homology to striatal neurons from mammals. Although the presence of primitive striatal components is evident in teleosts, there is no evidence of compartmentalization in the literature [16,17,19,20].

The telencephalic structures of the lungfish, *Protopterus annectens*, (a living species that represents an extinct member that gave rise to an amphibian), were immunohistochemically examined to study fish—amphibian transition [18,19]. A cluster of neurons positive for SP, ENK, TH [21], and NPY [22] were found in the basal telencephalon, revealing a primitive form of the basal ganglia composed of striatal and pallial subdivisions. A mosaic-like structure, however, was not described. TH-positive afferent terminals form a “dopamine island” that corresponds to the striosome during development of the mammalian striatum [23]. In African and Australian lungfish, a group of TH-expressing nerve terminals originating from mesencephalic cells was identified in the ventrolateral telencephalon. Double immunolabeling for ChAT, a matrix marker, and TH showed few ChAT-positive cells were present in the area rich with TH-positive neuropils [24]. It is plausible that cells with properties similar to those of matrix neurons were, if at all, only sparsely present, although there is the area analogous to the striosome that receives primitive nigrostriatal dopaminergic projections in lungfish.

The striatum constitutes a major part of the basal telencephalon in amphibians [25]. The striatal cells in *Xenopus laevis* expressed the transcription factor gene *Distal-less-4 of Xenopus* (*xDll4*, ortholog of mouse Dlx2) [26] in both embryonic and adult brains [25]. A transcription factor *Pax6*, which plays an important role in neuronal differentiation during mammalian development, was shown to be expressed in the *Xenopus* striatum [27], suggesting that the striatum in amphibians shares a part of its transcriptional profile with the developing mammalian striatum. An immunohistochemical study using antibodies against TH, SP, and ENK in *Rana perezi* showed that there were clearly demarcated areas labeled with these markers in the basal telencephalon [28]. Any compartment structures, however, were not reported. Another study in *Rhinella arenarum* tadpoles, showed that immunoreactivity for NPY, a matrix marker, was present in the developing striatum but decreased in number and staining intensity after metamorphosis into adult forms [29]. We found no evidence of compartmentalization in the amphibian striatum.

## 3. The Striatum in Non-Mammalian Amniotes

Fernandez et al. [30] studied expression patterns of Emx, Dlx, and Pax family homeobox genes in mice, chicks, turtles, and frogs to elucidate whether telencephalic subdivisions are phylogenetically conserved. They showed that the developing telencephalon can be categorized into pallial, intermediate, and striatal domains. The striatum developed in the basal telencephalon and was phylogenetically conserved throughout these different vertebrates, although the fate of intermediate domains differed between species.

### 3.1. Reptiles

A number of comparative anatomical studies have demonstrated the presence of homologies between reptilian and mammalian striata in their structural organization, connectivity, and neurochemical properties [9,11,31]. The expression of NPY, which is confined to the matrix in mammalian brains, was found only in sparsely scattered cells distributed in the striatum of the chameleon [32]. Immunohistochemical studies performed in the reptile *Caiman crocodilus* showed that the ventrolateral telencephalon, a homolog of the striatum, was divided into two distinct “small-celled” and “large-celled” fields [33,34]. SP-positive neurons [34], cholinesterase activity, and ascending catecholaminergic axon terminals derived from the midbrain tegmentum [33] were found in the small-cell field. In contrast, the large-cell field had far less catecholamine activity [33]. These studies revealed neurochemical properties preferentially seen in the striatum: elevated acetylcholinesterase activities showing the presence of cholinergic neurons and abundant dopaminergic terminals showing the presence of dopaminergic inputs from the SNc or the ventral tegmental area (VTA). In the turtle, *Chrysemys scripta*, SP-positive neurons were preferentially found in the small-cell field [34]. However, these papers did not mention if the cell fields were arranged in a mosaic fashion as in mammals. In the lizard, *Anolis carolinensis,* the gene expression of serotonin (5-HT) receptors was not confined to a mosaic pattern although expression patterns of 5-HT receptor subdivisions in the telencephalon resembled those reported in mammals [35]. While cell populations homologous to the striosome and matrix neurons were present in the reptilian striatum, the mosaic-like compartment organization of two groups of neurons was absent.

### 3.2. Birds

The avian striatum is divided into three components called the medial striatum, lateral striatum, and accumbens [36]. These structures have not been definitively related to any one specific part of the mammalian striatum, although they share some neurochemical expression patterns and connectivity to other parts of the telencephalon [36,37].

The expression of matrix markers was reported in the avian striatum. CB, a calcium-binding protein preferentially expressed in the matrix compartment in mammalian striatum, was found in neurons of the striatum of male zebra finch [38]. A differential distribution of calbindin-positive neurons was shown between the vocal-learner bird *Melopsittacus undulates*, and the non-vocal learner bird, *Colinus virginianus*, in which heavily-labeled cells were scattered amidst the weakly-labeled cells [39]. The calbindin-positive and negative neurons appeared to be intermingled. Bruce et al. [37] studied 12 different marker proteins enriched in the basal ganglia of the pigeon, *Columbia livia*. They showed a gradient of staining intensity of NPY and ENK, both of which are matrix markers, in the rostro-caudal and ventro-dorsal axis of the striatum, and found the globus pallidus “interwoven” into the striatum with finger-like structures in the rostral sections. The low levels of calbindin and high levels of parvalbumin in the medial part of the lateral striatum in pigeons resembled those in the matrix of the mammalian caudo-putamen [40].

The expression of striosome markers was also reported in the literature. The low calbindin and parvalbumin, and high SP levels in the medial part of the striatum raise the possibility that the area corresponded to the striosome compartment [41,42]. The expression of two types of dopamine receptors, D1A and D1B, was studied in the forebrain of the chick, *Gallus domesticus* [43]. The medial part of the striatum was visualized as a densely-packed and uniformly-stained nucleus positive for D1A receptors. Glutamate receptors are implicated in learned vocalization in the avian brain [44,45]. The mosaic pattern of GluR1 expression was not reported in the striatum of the songbird, *Taeniopygia guttata* [45] whereas it was one of the markers of the striosome in primates [46]. However, a clear segregation of the striosome within the matrix compartment was not obvious in avian striata, as described in earlier reports [8,47,48,49]. They suggested that neurons homologous to the striosome are homogenously distributed in the striatum.

## 4. Striatal Compartmentalization in the Mammalian Brain

The compartment structure of the striatum is considered to be evolutionarily conserved in mammals including humans [39,50]. Neurons forming these two compartments are generated during partly overlapping stages of striatal development [51,52,53]. In mice, striosomal neurons born earlier than embryonic day 13.5 (E13.5) (E10.5–11.5 in the caudal- and E12.5–13.5 in the rostral part) [53] form the striatal primordium in the basal part of the telencephalon. Then, the massive wave of later-generated matrix neurons migrates into the primordium and divide striosomal neurons into clusters [54]. Consequently, striosomal neurons form patchy cell clusters whereas matrix neurons occupy the space in between the striosomes, resulting in labyrinthine structures in a 3D space [2,55]. TH, an enzyme that generates dihydroxyphenylalanine, is rich in the so-called “dopamine islands”, which correspond to the striosomes, during development, whereas TH-immunoreactivity is slightly preferential to the matrix in adulthood. Both D1- and D2-receptor expressing medium spiny neurons are present either in the striosome or matrix [56]. The striatal compartment begins to appear perinatally and exhibits the mature pattern two weeks after birth in rodents [57]. Hagimoto et al. [58] identified differentially labeled progenitor cells in the striosome and matrix and cultured them in vitro. They showed that striosome cells were early-born, stationary, and mutually attractive in migratory behaviors whereas matrix cells were late-born, actively motile, and exhibited repulsive action against striosomal cells in later stages of striatal development. Their findings were consistent with a proposed model of compartmentalization of the striatum [58,59,60,61,62]. Studies using transgenic mice that selectively express enhanced green fluorescent protein (eGFP) either in the striosome or in the matrix, demonstrated distinct functions of these two compartments [56,63,64]. The establishment of mosaic architectures in the striatum is thus an outcome of precise temporal regulation and termination of neuronal migration during development.

In the mammalian striatum, two types of projection neurons can be identified based on different neurochemical and functional properties. Neurons expressing SP project to the GPi and SNc. This is called the direct pathway and is involved in promoting planned movement. Neurons expressing ENK project to the GPe. This is a part of the indirect pathway, which finally projects to the GPi via a projection to the subthalamic nucleus. Striatal projection neurons are derived from the lateral ganglionic eminence (LGE) [65,66,67]. The precursors of interneurons are reported to become postmitotic in the medial ganglionic eminence (MGE), tangentially migrate to a lateral position, and intermingle with projection neuron precursors to generate the striatal primordium, where they differentiate into cholinergic-, calretinin-positive-, or parvalbumin-positive interneurons [68]. Overexpression of Nkx2.1, a transcription factor expressed in the developing MGE, enhances the migration of interneurons to the striatum and reduces migration to the cortex [69], suggesting that Nkx2.1 is involved in the specification of striatal interneurons. Reports show that specific guidance cues consisting of diffusible molecules are involved in the radial migration of projection neurons [70] and tangential migration of interneurons [71,72,73] in the developing striatum. For example, netrin-1, which is secreted from the ventricular zone of the LGE, repels neuronal progenitors derived from the LGE and thereby promotes radial migration and differentiation of striatal projection neurons [70]. The tangential migration of striatal interneurons from the MGE was shown to be regulated by a combination of Nrg1/ErbB4 chemoattraction and EphB/ephrinB chemorepulsion [73]. Recent findings in mice showed that aberrant compartmentalization of the striatum induced by valproic acid administration during striosomal neurogenesis resulted in autism spectrum disorder-like phenotypes [74].

A traditional view is that the striatum can be subdivided into three functionally distinct parts called the sensori-motor, associative, and limbic domains [75], approximately located in the dorso-lateral, dorso-medial, and ventral striatum [76]. In addition to organization in the coronal plane, recent studies analyzing 3-dimentional structures have demonstrated extreme heterogeneity in the rostro-caudal axis [75,77,78]. Hunnicutt et al. [75] generated a comprehensive map to demonstrate cortico-striatal and thalamo-striatal input patterns to the striatum. Patterns of connectivity were consistent with the distribution of the three striatal domains; in addition, the authors showed the presence of another domain located in the most caudal part of the striatum which received strong inputs form the auditory and visual cortices. Gangarossa et al. [77] reported a specific region exclusively distributed with dopamine D1 receptor-expressing striatal projection neurons, which represent striato-nigral neurons forming the direct pathway, in the caudal part of the striatum. This part was shown to be dominated with matrix compartments, since neurochemical studies showed the expression of calbindin, VGluT1 and 2, and the lack of MOB and ENK. Miyamoto et al. [78] constructed a 3-dimensional map of neurochemical markers for the striosome and matrix and found a specific domain in the most caudal part of the striatum. The domain was a striosome-free space, exhibited a tri-laminar structure and selectively innervated from the motor and sensory areas in the neocortex whereas the striosome-rich more rostral part received inputs from the associative or limbic cortices. The most caudal part of the striatum appears to have unique properties, particularly in the lack of compartment structures. A difference of functional domains in the rostro-caudal axis is present in the human striatum, in which the anterior part (i.e., caudate nucleus) has associative and limbic functions and the posterior part (i.e., putamen) is involved in the execution of voluntary movements [1]. However, which part in the human striatum is comparable to the above-mentioned most caudal subdomain in rodents is unknown, and whether unique properties including functional connectivity of this domain are phylogenetically preserved has yet to be determined. Phylogenetic consideration in this review is mainly focused on the compartment structure in the dorsal striatum (caudo-putamen) because anatomical and functional integration of two compartments in the ventral striatum (nucleus accumbens and a part of the olfactory tubercle) was shown to be different from that in the dorsal striatum: For example, dopaminergic afferents from the substantia nigra or the VTA are differentially distributed in striosome and matrix compartments between the dorsal and ventral striatum [64,79,80]

Johnston et al. [54] performed a histological study in serial sections from rat, monkey and human striatum and found a remarkable increase in the striatal volume (>90 times) from rodents to humans. The authors also showed that the number of striosome was, in contrast, conserved among these species. Although a small difference in the proportion occupied by the striosome can be found across species (10.8%, 13.0% and 18.2%, rats, monkeys and humans) and also within species in the rostro-caudal axis of the striatum [54,81], a relatively constant ratio of 15% striosome and 85% matrix appears to be maintained in mammalian striatum [54]. We posit that the fundamental structure of the striatal compartment is conserved even after the evolution progress in the mammalian brain.

There have been reports suggesting that different thalamic nuclei send thalamo-striatal axons differentially to the striosome and matrix in the mammalian brain [1]. The paraventricular nucleus, one of the midline group of thalamic nuclei with strong connectivity to the limbic part of the brain, sends axons preferentially to the striosome [82]. By contrast, the parafascicular nucleus of the thalamus, which has connectivity to the sensorimotor areas in the neocortex [83,84], sends axons to the cholinergic neurons in the matrix. The pattern of connectivity would represent differential involvement of striosome and matrix compartments in limbic and sensorimotor parts of basal ganglia- thalamo-cortical circuits, respectively [1]. The axon terminals of the thalamo-striatal projection were shown to be less abundant compared with those of the cortico-striatal projection [85] when these two types of axon terminals were distinguished by immunolabeling with vesicular glutamate transporters (VGluT), VGluT1 and VGluT2, respectively [86,87]. In the mammalian brain, specific sensory information in the thalamus is forwarded to the cortex and then reaches the striatum after a dramatic increase in the number and complexity of contents [9] whereas direct projections from specific sensory thalamic nuclei are the main inputs to the striatum in amphibians [88]. Thus, the progressive involvement of the cortex in processing sensory information was the major evolutionary trend in the evolution of the basal ganglia-thalamo-cortical circuit [9]. The main focus of the next section is on the cortico-striatal projection in relation to the formation of striatal compartmentalization.

## 5. Emergence of the Six-Layered Cortex and the Striatal Compartment

The neocortex is considered to first appear as a uniform, six-layered sheet consisting of radially deployed neurons in the early small mammals that evolved from their reptilian ancestors during the transition of the Triassic/Jurassic periods [89] (for review see Figure 1). In amphibian brains, a six-layered cortex is absent [10]. Previous tract-tracing studies did not support the existence of massive connections between the pallium and the striatum [9], suggesting that the majority of information from sensory organs such as the eyes, ears, or skin is relayed to the thalamus and projected directly to the striatum without involvement of the pallial circuit [90]. This may reflect a more primitive repertoire of behaviors and movements in anamniotes [8]. After the anamniote-amniote transition, reptilian brains developed an elaborate projection from the pallium to the striatum [8,9], although the six-layered cortex was still lacking. In the reptilian-mammalian transition, a radical evolution occurred in the pallium, which acquired laminar organization resulting in the development of the isocortex, and was progressively involved in processing sensory inputs from the thalamus [9]. The mammalian cortex is parcellated into areas functionally specialized for processing multiple sensory modalities; these include the visual, auditory, and somatosensory systems [91,92]. All the cortical areas are known to send efferent projections to the striatum [93] and the cortico-striatal pathway thereby became a major input to the basal ganglia circuit. Lee and colleagues [90] suggest that involvement of the cortex in sensory-motor integration and development of the dense flow of information from the cortex to the striatum is one of the major advances in mammalian brains to gain more sophisticated cognitive functions and behaviors. We suggest that emergence of the cortex upstream of the sensory-motor processing system had a great impact on striatal structures.

Brain size expanded dramatically during mammalian evolution [94,95,96]. What happened to the size of each component? We plotted the fractional volume of the neocortex and the striatum as an exponential function of the total brain volume (Figure 2) according to the data presented by Stephan et al. [94], who measured the volume of each brain structure in 76 species from insectivorous mammals to human. We found that the striatal fraction is relatively constant (0.049 ± 0.009) whereas the neocortical fraction exhibited remarkable expansion from 0.1 in insectivorous mammals to 0.8 in human (Figure 2) [95]. Using the same data, we performed a nonlinear fit that revealed that the relationship between the cortex and the striatum was well described by a power law with an exponent as previously described in the thalamo-cortical volume relationship [96]. The striatal volume has expanded as a constant fraction within the brain as it underwent exponential expansion during mammalian evolution. The neocortex has progressively occupied a larger part of the brain, indicating that the volume expansion of the neocortex was much more explosive compared to the striatum. Given that virtually all cortical areas send axons to the striatum, the convergence of information through the cortico-striatal projection may have been exponentially enhanced during the course of mammalian evolution.

There are known to be two different kinds of cortico-striatal projection neurons depending on their clearly distinctive patterns of targeting [97,98]. Intra-telencephalic (IT) neurons send axons to the ipsi- and contralateral striatum and cortex but not to extra-telencephalic nuclei such as the thalamus or brainstem [97]. Pyramidal tract (PT) neurons project to the brain stem or, in some instances, directly to the spinal cord with collaterals to the ipsilateral cortex and subcortical nuclei including the striatum, thalamus, and superior colliculus; however, their axons do not cross the midline to the contralateral hemisphere. Hooks et al. [98] performed 3-dimensional reconstruction analysis and found important differences in projection topography between these two types of projections. The IT-type cortico-striatal projection had a more spread target in the striatum and was substantially overlapped between its targets from corresponding cortical areas whereas the PT-type projection had a more focal target area. The authors suggest that more localized targeting of PT-type projection might be useful for activation of a specific subset of basal ganglia circuitry involved in motor control. In contrast, the IT-type projection might represent a broader signal for coordination of movement.

Accumulating evidence shows that late-born matrix neurons receive their dominant input from the neocortex, whereas early-born striosomal neurons connect with components of limbic circuits such as the amygdala [1,55,99,100,101,102] (Figure 3). Neuroanatomical studies deploying anterograde tracers demonstrated that early-born neurons in deep cortical layers projected primarily to the striosome, and late-born neurons in the superficial layers projected primarily to the matrix [55,103,104] The matrix compartment receives inputs from phylogenetically younger neocortical areas such as the somatosensory cortex, in which upper parts of layer V are well-defined. The striosome, in contrast, receives inputs from the limbic cortex, in which the superficial layer is less-developed [1]. The region-specific cortical layer formation is consistent with compartment specific targeting of cortico-striatal projections as shown in Figure 3. These findings suggest that the cortico-striatal connectivity is formed in a phylogenetically conserved manner. Using a combination of birthdating and axon-tracing analyses, the cortico-striatal projection neurons were shown to be born between E12.5 and E14.5 and elongated cortico-striatal axons reached the striatum perinatally [105], when the striatal compartment began to appear, and later exhibited the mature pattern at two weeks, postnatally [57]. Thus, in the ontogeny of the mammalian brain, striatal compartmentalization develops along with development of the cortico-striatal connectivity. We propose that the emergence of striatal compartmentalization is concordant with elaboration of the six-layered cortex and development of the cortico-striatal connectivity in phylogeny, which contributed to the gain of more complex and sophisticated brain functions to survive in an ecosystem.

## 6. Conclusions

Recent reports show that there is considerable similarity between neuronal populations that form the striatum among vertebrates, including—to some extent—anamniotes with regard to neurotransmitter contents, physiology, and connectivity. However, the clear compartmentalization in the striatum exclusively develops in the mammalian brain, which has a six-layered cerebral cortex. During the evolution of mammals, the number of cortical neurons and corticofugal fibers increased extensively and the striatum subsequently received massive inputs from the cerebral cortex. By gathering information from the external world via sensory organs such as the eyes, nose and ears, the brain maps multiple streams of information onto a single axis of value [90,106] and produces behaviors for survival. The development of higher intelligence in phylogeny must be mainly caused by the emergence of exponentially-expanded information processing in the cortex, the striatum, and other structures in the telencephalon. Striatal compartmentalization in tandem with increased information processing via the cortico-striatal pathway may have contributed to the evolution of complex decision-making and enhanced behavioral flexibility in animals with a highly developed cortex such as primates and humans.

## Figures and Tables

**Figure 1 brainsci-09-00090-f001:**
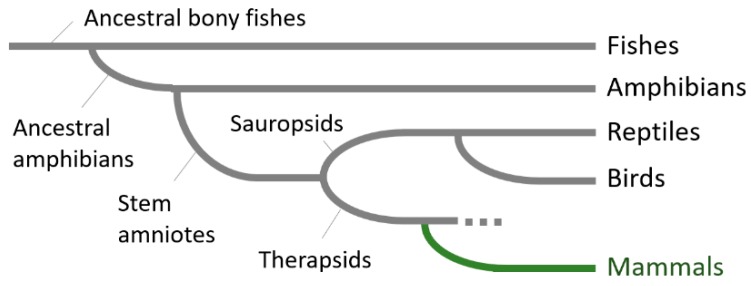
A phylogenetic tree of vertebrate evolution. A diagram showing the lines of evolutionary descent of different vertebrates from a common ancestor. The ancestral bony fish gave rise to ancestral amphibians. Through water-land transition, ancestral amphibians gave rise to stem anamniotes, which then separately evolved into sauropsids and therapsids. Sauropsids are not only the ancestors of existing reptiles but also gave rise to the ancestors of birds. Mammalian species have evolved from therapsid ancestors, although many groups of the therapsids have become extinct (dashed line). The six-layered cortex was inherited by the mammalian ancestor of therapsids (green line) more than 200 million years ago [10]. Modified from Jarvis et al. [11].

**Figure 2 brainsci-09-00090-f002:**
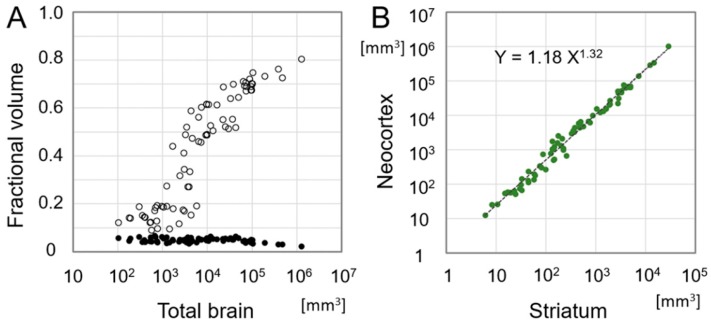
Evolutionary scaling relation between the striatum and neocortex. (**A**) A plot showing fractional volumes of the striatum (black dots) and the neocortex (open circles) in the total brain volume, which is defined as 1. The data are according to measurements of 76 species from insectivorous mammals to human [94]. Note that the striatal fraction is relatively constant throughout mammalian evolution whereas the neocortical fraction increased explosively. (**B**) A plot showing the volume of the neocortex as a function of the volume of the striatum according to the same data shown in (**A**). The best-fit exponential formula is: Y = 1.18X^1.32^, where X and Y are the volume of the striatum and the neocortex in cubic millimeters, respectively.

**Figure 3 brainsci-09-00090-f003:**
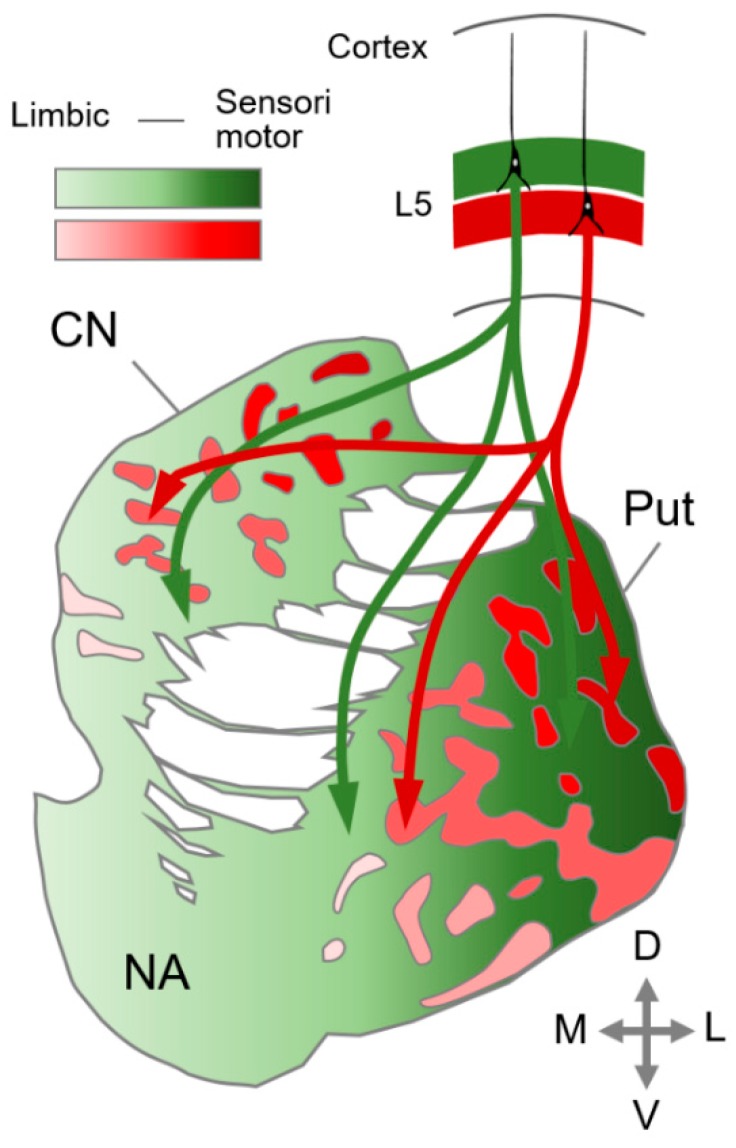
Segregated connectivity of the cortico-striatal pathway. The cortico-striatal neurons in the deeper (red) and upper (green) parts of cortical layer 5 (L5) project preferentially to the striosome (red) and matrix (green) compartments, respectively. The gradient of colors indicates polarized distribution of cortico-striatal connectivity. The light red and light green indicate parts of the striatum that receive inputs predominantly from the limbic cortex. The dark red and dark green indicate parts of the striatum that receive inputs predominantly from the sensori-motor cortex. Abbreviations: CN, caudate nucleus; NA, nucleus accumbens; Put, putamen. Orientations: the lateral (L) is to the right, the dorsal (D) is to the top.

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
