# Peer review of "Parallel Emergence of a Compartmentalized Striatum with the Phylogenetic Development of the Cerebral Cortex"

_brainsci, 2019, doi:10.3390/brainsci9040090_

Reviewer 1 Report

The submitted manuscript is a brief review of development and comparative anatomy of the dorsal striatal compartments, termed striosome/patch and matrix, among vertebrates.

This review broadly covers development of the striatum from fish to humans and focuses on neurochemical markers and anatomical circuitry as the basis for evaluating the presence of the striosome and matrix compartments of the dorsal striatum. However, isn't it true that the ventral striatum is an older structure with similar connectivity? It would be useful if the authors explained why they did not instead compare aspects of the ventral striatum across distantly related species.

Figure 2, in which the authors plot data from a previous publication, is confusing. The graph in panel A shows that the proportion of neocortical volume out of the whole brain volume increases as a function of whole brain volume whereas striatal volume remains constant. However, the graph in panel B shows that there is a nearly perfectly linear and positive relationship between the volumes of neocortex and striatum across species. Is the inconsistency because the striatal volume/whole brain volume in (A) appears low because of the rapidly increasing neocortical volume in the denominator? If yes, then a comparison of the neocortex and striatum to the same other brain structures would be expected to show that the proportion of neocortical and striatal volume both increase relative to other structures as brain size increases, as expected from panel B.

As a conclusion, the authors propose that the striatal compartments co-evolved with a layered cortex because it was shown that the striosomes and matrix are differentially innervated by older and newer layers, respectively. However, they provide only one older reference for this finding that is key to their conclusion- are there more references that support or dispute this original finding? Also, is this finding confounded by their being differential projections from upper and lower layer 5 in sensorimotor (matrix-preferring) vs. limbic cortices? Please discuss.

Minor corrections:

Line 40: Missing verb ('exist' could be added after 'matrix,'.

Line 73: Replace 'vertebrate' with 'animal' - AchE and TH exist in animals that are phylogenetically older than vertebrates.

Line 90: Missing verb ('suggested' could be added after 'populations'.

Line 91 and Line 175: Replace mammalian with mammals.

Line 162: Add 'of' after 'expression'.

Line 137: Why do the authors think that ascending dopamine fibers in reptiles are from the homolog of the SNc and not the homolog of the VTA? In mammals, both the VTA and the SNc innervate the dorsal striatum - although there are more inputs from the SNc, the VTA is a phylogenetically older region per my rudimentary understanding.

Line 176: Delete 'non-'. The birthdates of striosome and matrix neurons do clearly overlap to a significant degree.

Line 183-184: Change 'is preferentially found in the matrix' to 'is slightly preferential to the matrix'.

End of Line 185: Typo, a word seems to be missing.

Line 284: Typo, 'progresses' is used as a noun but it is a verb.

Author Response

Reviewer #1

Comment #1: This review broadly covers development of the striatum from fish to humans and focuses on neurochemical markers and anatomical circuitry as the basis for evaluating the presence of the striosome and matrix compartments of the dorsal striatum. However, isn't it true that the ventral striatum is an older structure with similar connectivity? It would be useful if the authors explained why they did not instead compare aspects of the ventral striatum across distantly related species.

RESPONSE: According to the reviewer’s comment, we briefly mentioned the difference of anatomical and functional integration in two compartments between the dorsal (caudo-putamen) and the ventral (nucleus accumbens and a part of olfactory tubercle) striatum at lines 241-247 in the revised version.

Comment #2: Figure 2, in which the authors plot data from a previous publication, is confusing. The graph in panel A shows that the proportion of neocortical volume out of the whole brain volume increases as a function of whole brain volume whereas striatal volume remains constant. However, the graph in panel B shows that there is a nearly perfectly linear and positive relationship between the volumes of neocortex and striatum across species. Is the inconsistency because the striatal volume/whole brain volume in (A) appears low because of the rapidly increasing neocortical volume in the denominator? If yes, then a comparison of the neocortex and striatum to the same other brain structures would be expected to show that the proportion of neocortical and striatal volume both increase relative to other structures as brain size increases, as expected from panel B.

RESPONSE: According to the reviewer’s comment, we add sentences to more clearly explain an exponential scaling relation between the striatal- and neocortical volumes at lines 303-306 in the revised version. We would kike to point out that the Y axis in Figure 2B was drawn as an exponent. If the same data are plotted in the graph with the linear Y axis (shown on the right), you will find that the volume expansion of the neocortex as a function of the striatal volume can be expressed as an exponential formula as shown in Figure 2B.

After careful consideration, we decided to leave Figure 2B as the original version in align with the previous study on evolutionary scaling law by Stevens (2001).

Comment #3: As a conclusion, the authors propose that the striatal compartments co-evolved with a layered cortex because it was shown that the striosomes and matrix are differentially innervated by older and newer layers, respectively. However, they provide only one older reference for this finding that is key to their conclusion- are there more references that support or dispute this original finding? Also, is this finding confounded by their being differential projections from upper and lower layer 5 in sensorimotor (matrix-preferring) vs. limbic cortices? Please discuss.

RESPONSE: According to the reviewer’s comment, we added sentences to support compartment specific targeting of cortico-striatal projections shown in Figure 3 at lines 337-341 of the revised version. Appropriate references were additionally included at lines 336-337 in the revised version.

Minor corrections:

Line 40: Missing verb ('exist' could be added after 'matrix,'.

Line 73: Replace 'vertebrate' with 'animal' - AchE and TH exist in animals that are phylogenetically older than vertebrates.

Line 90: Missing verb ('suggested' could be added after 'populations'.

Line 91 and Line 175: Replace mammalian with mammals.

Line 162: Add 'of' after 'expression'.

Line 137: Why do the authors think that ascending dopamine fibers in reptiles are from the homolog of the SNc and not the homolog of the VTA? In mammals, both the VTA and the SNc innervate the dorsal striatum - although there are more inputs from the SNc, the VTA is a phylogenetically older region per my rudimentary understanding.

Line 176: Delete 'non-'. The birthdates of striosome and matrix neurons do clearly overlap to a significant degree.

Line 183-184: Change 'is preferentially found in the matrix' to 'is slightly preferential to the matrix'.

End of Line 185: Typo, a word seems to be missing.

Line 284: Typo, 'progresses' is used as a noun but it is a verb.

RESPONSE: We appreciate the corrections indicated by the reviewer. All of them have been changed appropriately in the revised version.

Reviewer 2 Report

This is a very interesting and unique review covering evolutionary aspects of striatum structures. I do not have any specific concern regarding the contents. I just wonder whether the authors can include one additional schematic diagram for the striatal architectures of non-vertebrate animals. 

Author Response

Reviewer #2

Comment #1: This is a very interesting and unique review covering evolutionary aspects of striatum structures. I do not have any specific concern regarding the contents. I just wonder whether the authors can include one additional schematic diagram for the striatal architectures of non-vertebrate animals.

RESPONSE: We appreciate the reviewer’s positive comment on your work. Concerning about the schematic diagram, we have referred to two extensive review papers by Reiner et al. (1998) and Smeets et al. (2000) at lines 60-61 of the revised version. They showed beautiful drawings about evolutional changes in striatal architectures.